cancer; precision oncology; genomics; tumour heterogeneity; proteomics

**Corresponding author:**
Jacek Jassem;
Email: jjassem@gumed.edu.pl

# Heterogeneity in precision oncology

Bartłomiej Tomasik[1], Filip Garbicz[2,3,4] [ID], Marcin Braun[5], Michał Bieńkowski[6] and Jacek Jassem[1] [ID]

[1]Department of Oncology and Radiotherapy, Faculty of Medicine, Medical University of Gdańsk, Gdańsk, Poland; [2]Department of Pathology, Dana-Farber Cancer Institute, Boston, MA, USA; [3]Department of Experimental Haematology, Institute of Haematology and Transfusion Medicine, Warsaw, Poland; [4]Department of Immunology, Medical University of Warsaw, Warsaw, Poland; [5]Department of Pathology, Chair of Oncology, Medical University of Łódź, Łódź, Poland and [6]Department of Pathomorphology, Medical University of Gdańsk, Gdańsk, Poland

## Abstract

Precision oncology is a rapidly evolving concept that holds great promise in cancer treatment. However, a cancer complexity attributed to genomic and acquired tumour heterogeneity limits treatment effectiveness and increases toxicity. These limitations refer to both systemic therapies and radiotherapy, which are two mainstays of non-invasive cancer treatment. By understanding cancer heterogeneity and utilising advanced tools to personalise treatment strategies, precision oncology has the potential to revolutionise cancer care. In this article, we review the current status of precision oncology in solid tumours, specifically focusing on the impact of tumour heterogeneity and genomic patient features on systemic therapies and radiation. We also discuss the implementation of novel tools, such as next-generation sequencing and liquid biopsies, to overcome this problem.

## Impact statement

Precision oncology, one of the most promising applications of precision medicine, uses molecular and genetic information to customise cancer treatments, considering the individual characteristics of each patient's tumour. To further advance the field, precision oncology increasingly incorporates knowledge of cancer heterogeneity, on both spatial and temporal levels. Addressing these complexities with modern precision radiotherapy and systemic therapies is the key to targeting all cancer cell subpopulations. The future vision of precision oncology involves continuous advancements in technological and analytical methods, leading to further treatment personalisation. This progress will ultimately contribute to a paradigm shift in cancer care to improve patient outcomes significantly. Access to advanced tools should be improved in terms of availability and affordability while addressing the need for routine genomic profiling across various regions of primary and metastatic tumours to understand cancer heterogeneity comprehensively.

## Introduction

Precision medicine is a novel approach to treatment and prevention that tailors strategies to the unique characteristics of individual patients, including their genetics, environment and lifestyle. It differs from conventional evidence-based medicine, which generally relies on average clinical benefits in the studied populations (Tonelli and Shirts, 2017; Blackstone, 2019). Precision medicine is supported by advances in technology and medical research, such as using genomic sequencing and big data analysis to identify individualised treatment options.

The decision-making process in precision medicine is based on predictive biomarkers, which offer insights into the underlying molecular mechanisms of tumorigenesis and allow the identification of potential therapeutic targets. In clinical settings, biomarkers can predict which patients are most likely to benefit from specific therapies, optimise treatment efficacy and reduce toxicity. Further, biomarkers enable early cancer detection and treatment monitoring, thereby increasing its efficacy. In essence, biomarkers are transformative tools of personalised medicine, driving more accurate, effective and safer cancer treatments (Slikker, 2018).

The two most commonly used markers are prognostic and predictive biomarkers. A prognostic biomarker is a clinical or biological indicator that offers insights into the probable health outcome of an individual patient, such as disease recurrence or death, regardless of the treatment pursued. In turn, a predictive biomarker signifies the potential advantage to the patient, resulting from a specific treatment (Sechidis et al., 2018). Other biomarkers include predisposing biomarkers, indicating the potential for developing a disease (Califf, 2018) and pharmacogenomic biomarkers, informing about the drug efficacy and toxicity based on the underlying genetic composition (Lauschke et al., 2017). The United States Food and Drug Administration and the

National Institutes of Health published the Biomarkers, EndpointS and other Tools (BEST) resource, describing the extensive list of biomarkers used in translational science (Cagney et al., 2018).

Precision oncology is a concept that customises oncological care based on unique patient genomics and clinical, genetic, proteomic, transcriptomic or phenotypic tumour features (de and Ashworth, 2010; Collins and Varmus, 2015). Precision oncology has achieved unprecedented advancements through rigorous scientific evidence and extensive computational analyses (Mirnezami et al., 2012). However, challenges such as accurate data interpretation, precise patient stratification and the development of successful targeted therapies for specific genomic aberrations require further efforts (Prasad et al., 2016).

To overcome these obstacles, precision oncology requires innovative clinical trial designs that account for patient and tumour heterogeneity and the dynamic nature of cancer evolution (Chen and Snyder, 2013). Integrating precision oncology into clinical practice is a key goal of the Precision Medicine Initiative, which was launched by the US government in 2015.

In the present article, we discuss the impact of tumour and patient heterogeneity on treatment outcomes in solid tumours oncology and explore how precision systemic therapies and radiotherapy can mitigate these obstacles. The analysis will focus on scrutinising pivotal studies, such as the Molecularly Aided Stratification for Tumour Eradication Research (MASTER) (Horak et al., 2017), the National Cancer Institute Molecular Analysis for Therapy Choice (NCI-MATCH) trial (Flaherty et al., 2020 and other pertinent research, to better understand customised cancer therapy. We also present current investigative endeavours and interdisciplinary collaborations to optimise cancer therapy in all patients, regardless of their genetic makeup (Topol, 2014; Jameson and Longo, 2015; Figure 1). The examples provided here should be considered illustrative, as no comprehensive literature analysis on this topic was attempted.

## Systemic therapies

Systemic therapies, which involve drugs circulating throughout the body, are fundamental to cancer treatment (Chabner and Roberts, 2005). Precision oncology has revolutionised systemic therapies by better allocating standard chemotherapy and has paved the way for specific targeted therapies (Schwaederle et al., 2015). However, cancer heterogeneity, both spatial and temporal, highly impacts the effectiveness of these therapies (Greaves and Maley, 2012). As a result, one of the major challenges in oncology is customising systemic therapies for each patient and tumour characteristics (Leichsenring et al., 2019).

### *Tumour heterogeneity*

Whereas cytotoxic chemotherapy is essential for many malignancies, it is generally recognised as a one-size-fits-all approach, which may not be optimal for patients with genetically diverse tumours. Precision oncology considers tumour genetic heterogeneity, thus can improve the efficacy of standard treatments, identify druggable targets for specific tumours and select patients who are more likely to benefit from customised treatments (Massard et al., 2017).

The relationship between specific genomic alterations, genetic inter- and intratumour heterogeneity and the effectiveness of cancer treatment has been well established (Schwaederle et al., 2015; McGranahan and Swanton, 2017). Tumours with certain genetic

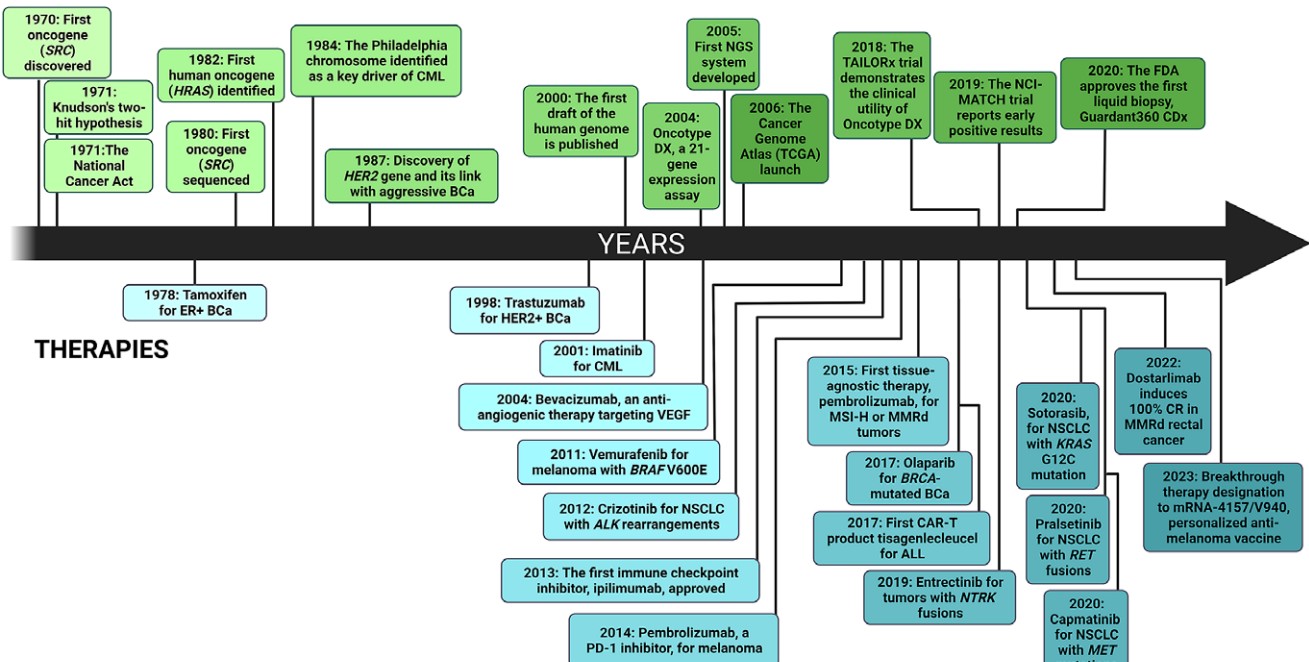

**Figure 1.** Timeline showing the highlights of clinical precision medicine. ALK, anaplastic lymphoma kinase; ALL, acute lymphoblastic leukaemia; BCa, breast carcinoma; BRAF, v-Raf murine sarcoma viral oncogene homologue B' BRCA, BReast CAncer gene; CAR-T, chimeric antigen receptor T-cell therapy; CML, chronic myeloid leukaemia; CR, complete response; ER+, oestrogen receptor-positive; HER2, human epidermal growth factor receptor-2; KRAS, Kirsten rat sarcoma virus; MET, hepatocyte growth factor receptor; MMRd, mismatch repair deficiency; MSI-H, high microsatellite instability; NGS, next generation sequencing; NSCLC, non-small cell lung cancer; PD-1, programmed death receptor-1; RET, Ret Proto-Oncogene; VEGF, vascular endothelial growth factor.

alterations differ in their susceptibility to classical cytotoxic chemotherapy. For example, mutations in the *TP53*, *KRAS*, *PTEN* or *RB1* genes are associated with resistance to chemotherapy (Custodio et al., 2009; Perrone et al., 2010), *BRCA1* and *BRCA2* mutations denote chemosensitivity to platinum compounds (Pennington et al., 2014), and *MGMT* methylation in glioblastoma is associated with a better response to temozolomide (Stupp et al., 2005).

Knowledge of genetic tumour heterogeneity has been extensively used in targeted cancer therapies (Figure 2). If druggable, genetic alterations are primarily used as therapeutic targets; however, many also serve as predictive markers for treatment effectiveness. In colorectal cancer, cetuximab, which is a chimeric antibody against the epidermal growth factor receptor (EGFR), is effective only against wild-type rat sarcoma (RAS) family oncogenes (Van Cutsem et al., 2009; Douillard et al., 2013). Conversely, in lung cancer, EGFR tyrosine kinase inhibitors are less effective in patients with coexisting TP53 (Aggarwal et al., 2018; Sun et al., 2023) or KRAS mutations (Massarelli et al., 2007), which can activate alternative signalling pathways bypassing the EGFR pathway. In breast cancer, human epidermal growth factor receptor-2 (HER2)

inhibitors are widely used to treat patients with HER2-overexpressing or HER2-amplified tumours, but they are less effective in patients with coexisting mutations in fibroblast growth factor receptor-1 (FGFR1) or receptor-2 (FGFR2) genes (Hanker et al., 2017). FGFR alterations correlate with resistance to several targeted and standard therapies across different malignancies (Babina and Turner, 2017), while the mechanistic and prognostic role of FGFR1–4 protein overexpression remains equivocal (Piasecka et al., 2019).

Some genetic alterations are druggable only in specific tumours, whereas others can be targeted across biologically and clinically different malignancies. The inhibitors of cyclin-dependent kinases 4 and 6 (CDK4/6) are effective and routinely administered to treat advanced hormone receptor-positive, HER2-negative breast cancer, though CDK4/6 alterations are not a hallmark of these cancers and do not predict the effectiveness of this therapy (Suski et al., 2021; Cristofanilli et al., 2022). Furthermore, CDK4/6 inhibitors are inefficient in liposarcomas harbouring the amplification of *CDK4/6* and murine double minute 2 (*MDM2*) genes (Sbaraglia et al., 2021). Other examples are ivosidenib, an isocitrate dehydrogenase-1

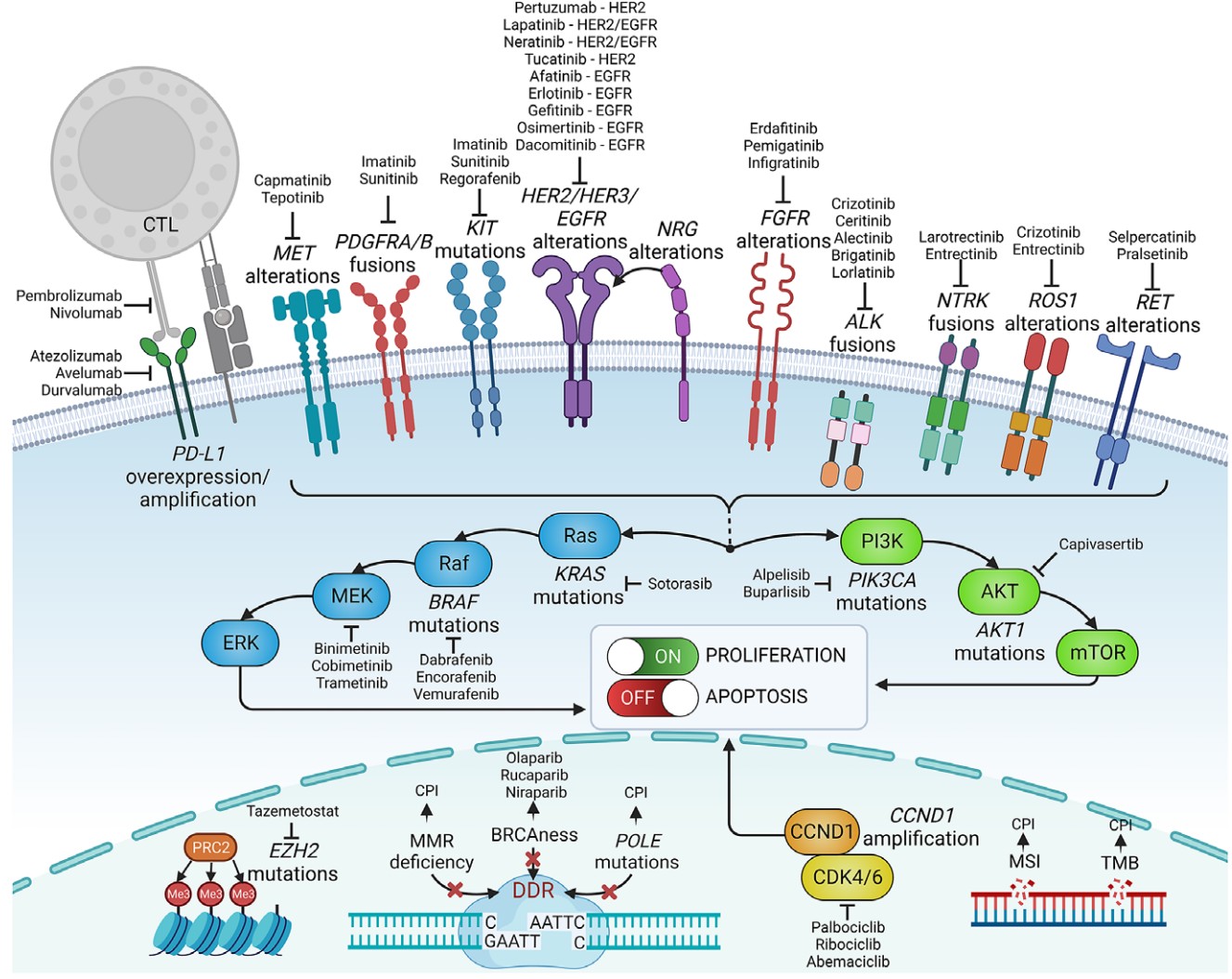

**Figure 2.** Examples of precision medicine biomarkers used in oncologic practice, together with respective targeted therapies approved in patients harbouring such lesions. CPI, checkpoint inhibitor; CTL, cytotoxic lymphocyte; DDR, DNA damage response; MMR, mismatch repair; MSI, microsatellite instability; PD-L1, programmed death-ligand 1; TMB, tumour mutational burden.

(IDH1) inhibitor, and enasidenib, an IDH2 inhibitor, which effectively targets relapsed or refractory acute myeloid leukaemia with IDH1/2 mutations (Cerchione et al., 2021) but that are not effective in gliomas bearing these mutations. In turn, some targeted therapies, for example, entrectinib, which targets neurotrophic tyrosine receptor kinase (*NTRK*) fusions, and ROS oncogene 1 (*ROS1*) rearrangements, are effective across different solid tumour types, including lung cancer, colorectal cancer and thyroid cancer (Doebele et al., 2020; Drilon et al., 2020). Similarly, V-raf murine sarcoma viral oncogene homologue B1 (BRAF) and mitogen-activated protein kinase (MEK) inhibitors were recently approved with a tumour-agnostic indication for unresectable or metastatic solid tumours harbouring the *BRAF V600E* mutation (Gouda and Subbiah, 2023).

Immune-oriented therapies, such as immune checkpoint inhibitors (ICIs) or chimeric antigen receptor T-cells (CAR-T), have revolutionised cancer treatment. However, correctly identifying good responders remains challenging. Genetic heterogeneity in tumours may elicit variable responses to ICIs. Malignancies with a high tumour mutational burden (TMB) and neoantigen load are more responsive to ICIs. Patients with high-TMB non-small cell lung cancer (NSCLC) or melanoma achieve significant improvements in survival with ICIs compared with those with low TMB (Ning et al., 2022; Ricciuti et al., 2022). However, intratumour or intersite (primary vs. metastatic foci) heterogeneity leading to spatial neoantigen expression variability might result in the escape of certain subclones from immune surveillance (McGranahan and Swanton, 2017). Different tumour types (e.g., colorectal or endometrial cancers) with microsatellite instability or mismatch repair deficiency are highly responsive to ICIs (Cercek et al., 2022; O'Malley et al., 2022), whereas tumours with some mutations may be ICI resistant. For instance, *EGFR*-mutant NSCLCs are less sensitive to ICIs than wild-type *EGFR* (Mazieres et al., 2019). Melanomas with overactive WNT/β-catenin signalling are less infiltrated by T-cells and, thus, less susceptible to ICIs (Spranger et al., 2015).

Cancers with high levels of intrinsic heterogeneity, which can be defined as the presence of different genetic clones within a single tumour, are usually less likely to benefit from chemotherapy and targeted therapies (McGranahan and Swanton, 2017) because of the presence of drug-resistant subclones within the tumour that can contribute to rapid relapse after the initial response to therapy (Burrell et al., 2013; Almendro et al., 2014). Computational modelling and in situ analyses have shown that genetic and phenotypic heterogeneity can greatly affect tumour evolution during chemotherapy and treatment outcomes (Almendro et al., 2014). Recent advances in genomics and single-cell sequencing have shed light on the molecular mechanisms underlying tumour heterogeneity, paving the way for the development of novel personalised therapeutic strategies (Dagogo-Jack and Shaw, 2018; Ramón et al., 2020; Labrie et al., 2022). Further development of precision medicine for systemic anticancer therapies heavily relies on better understanding and addressing intratumour heterogeneity. However, because of diagnostic limitations, intratumour heterogeneity cannot yet be routinely exploited in guiding treatment options. It is expected that circulating tumour DNA (ctDNA) and single-cell analysis techniques may enable a detailed characterisation of tumour cell populations and better inform personalised treatment strategies (Nath and Bild, 2021; Tivey et al., 2022). ctDNA dynamic profiling allows for real-time monitoring of tumour evolution and adapting treatment strategies as the tumour mutates and evolves. Recently, ctDNA-guided therapy was shown to be beneficial in patients with NSCLC and colorectal cancer (Jee et al., 2022; Tie et al., 2022).

Precision medicine approaches may considerably improve cancer treatment outcomes, provided that the complex interplay between tumour genetics and response to systemic treatment is better understood. Basic and translational studies are essential for identifying next-generation predictive biomarkers. Novel clinical trial designs, such as basket-type trials assessing the druggability of specific targets across different tumour types, and umbrella-type trials evaluating the efficacy of specific or various targeted therapies in specific cancer diagnoses (Figure 3 and Table 1), may prompt the development of new tailored therapies (Subbiah, 2023).

To compile the data presented in Tables 1 and 2 (please see below), we employed a multi-pronged search strategy. Firstly, a targeted search was conducted on PubMed using the following query: ("precision medicine"[Title/Abstract] OR "targeted therapy"[Title/Abstract] OR "personalised medicine"[Title/Abstract]) AND "clinical trial"[Publication Type] AND ("2013/01/01"[PDAT]: "2023/12/31"[PDAT]). This query was designed to yield articles classified as "clinical trials" focusing on "precision medicine," "targeted therapy," or "personalised medicine," and published between January 1, 2013, and December 31, 2023. In addition to PubMed, we supplemented the articles with data from the clinical trials registry ClinicalTrials.gov and information gathered from sessions at the European Society for Medical Oncology (ESMO) Meetings and American Society of Clinical Oncology (ASCO) Annual Meetings held between 2018 and 2023. These conferences are recognised platforms that regularly feature key developments in precision medicine trials in oncology.

### *Germline heterogeneity*

Response and adverse reactions to systemic therapies can vary substantially between individuals. This variability has been attributed mainly to the inherited genomic variants that inactivate protein-coding genes (Karczewski et al., 2020). The therapy may be impacted on several levels, including direct drug–target interactions, drug metabolism (including drug activation and removal) and downstream effects (e.g., DNA damage). Hence, considering patient pharmacogenomics can improve treatment outcomes, decrease toxicity and reduce costs.

The drug-metabolising enzyme known as cytochrome P450 2D6 (CYP2D6) is the most thoroughly examined and variable polymorphic enzyme (Zhou and Lauschke, 2022). Its deficiency is inherited through an autosomal recessive trait, and individuals carrying this alteration are classified as poor metabolisers. However, the remaining subjects (extensive metabolisers) display considerable variability in their enzymatic activity (Bertilsson et al., 2002). The gene encoding the CYP2D6 protein is highly polymorphic, with over 100 allelic variants described to date (Gaedigk et al., 2018). Specific genetic variations in the CYP2D6 gene can cause altered activity of the cytochrome P450 2D6 enzyme, which is involved in the metabolism of tamoxifen (a selective oestrogen modulator used to treat breast cancer). Individuals with reduced CYP2D6 activity (e.g., *4, *5 and *6 alleles) account for up to 10% of patients (Ingelman-Sundberg, 2004; Crews et al., 2014). These individuals have a lower efficacy of tamoxifen than those with normal activity (Lim et al., 2007; Schroth et al., 2007; Goetz et al., 2013). Likewise, the ultrarapid metabolisers (e.g., with *1xN or *2xN alleles, about 1–2% of patients) show lower endoxifen (a tamoxifen metabolite) concentrations and worse outcomes compared with patients with normal CYP2D6 activity (Wegman et al., 2005; Schroth et al., 2007; Crews et al., 2014).

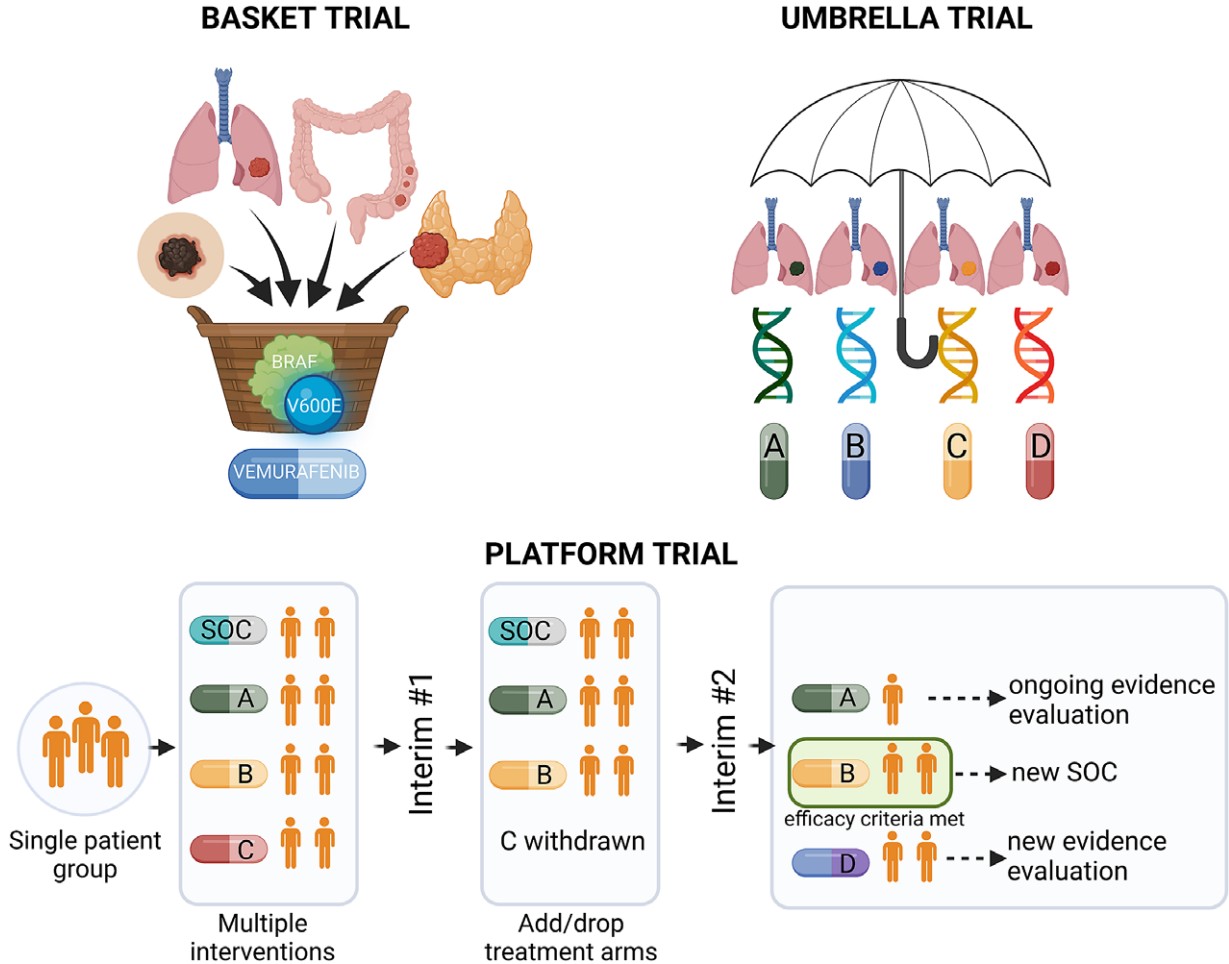

**Figure 3.** Types of precision medicine clinical trials. BRAF, v-Raf murine sarcoma viral oncogene homologue B; SOC, standard of care.

Similarly, variations in the DPYD gene (e.g., *2A or *13), which encodes the dihydropyrimidine dehydrogenase, an enzyme involved in the metabolism of fluoropyrimidines, such as 5-fluorouracil and capecitabine, can lead to reduced enzyme activity and an increased risk of severe toxicity (Amstutz et al., 2009; Offer et al., 2014; Henricks et al., 2018).

Finally, the UGT1A1 gene, which encodes for the uridine diphosphate glucuronosyltransferase 1A1 enzyme, is involved in the metabolism of irinotecan (a topoisomerase I inhibitor); variations reducing its activity (including *6, *27 and *28) may cause severe toxicity, including neutropenia, diarrhoea or infection (Iyer et al., 2002; Innocenti et al., 2004; Marcuello et al., 2004; Xu et al., 2016).

Genomic polymorphisms in drug targets may also affect interactions with drugs. Several HER2 gene variants have been shown to impact the effectiveness of trastuzumab (a monoclonal antibody targeting HER2) in HER2-positive breast cancer patients. The F117L variant, which is located in the extracellular domain of the HER2 protein, impairs trastuzumab binding by approximately threefold compared with wild-type HER2. Decreased binding affinity is attributed to the introduction of a leucine residue, which causes a steric hindrance and disrupts the protein conformation at the binding site (Gaborit et al., 2011). In turn, a variant within the HER2 intracellular kinase domain (T798I) leads to increased kinase activity, conferring resistance to lapatinib (Bose et al., 2013). Another example of EGFR polymorphism is R521K (rs2227983),

which may decrease the response to cetuximab in patients with metastatic colorectal cancer (Graziano et al., 2008). As a result, patients who carry this allele experience a lower incidence of skin toxicity during cetuximab treatment (Klinghammer et al., 2010; Fernández-Mateos et al., 2016).

Finally, variants affecting the efficiency of DNA damage repair may affect the response and toxicity of numerous drugs, including platinum agents, alkylating agents, topoisomerase II inhibitors, antimetabolites and poly-ADP ribose polymerase inhibitors. The rs3212986 variant (C8092A) of the ERCC1 gene, which is a component of the nucleotide excision repair pathway, is associated with a poor response to platinum agents in NSCLC and ovarian cancer (Zhou et al., 2004; Krivak et al., 2008). Similarly, the rs13181 (L751G) single nucleotide variant (SNV) in the XPD gene is associated with the poor efficacy of platinum agents in NSCLC (Park et al., 2001).

## Radiotherapy

Radiotherapy is a vital component of cancer treatment, applicable in around 60% of patients (Citrin, 2017). Until recently, radiotherapy was prescribed on the empirical basis of a one-fits-all approach, assuming a similar response to the same radiation dose. Recent advances in precision medicine have enabled the use of more

**Table 1.** Selected clinical trials investigating personalised cancer therapies

| Trial ID/Name | Patient population | Intervention | Precision technologies | Study design | Primary endpoint | N | Results |
|---|---|---|---|---|---|---|---|
| *MASTER* (Molecularly Aided Stratification for Tumour Eradication Research) (Horak et al., 2021) | Adults with advanced solid tumours (age < 51 years) and patients with rare tumours, including rare subtypes of more common entities, regardless of age (221 different ICD-O-3 codes), who exhausted curative treatment options | Evaluation of biomarkers' clinical actionability and assignment of molecularly informed therapies in semiweekly, multicentre MTB conferences | WES, WGS, RNA-seq | Multicentre, prospective observational study – master observational trial (Dickson et al., 2020) | PFSr | 1,310 | Of 300 patients evaluable for PFSr, 107 (*35.7%*) had a PFSr >1.3 |
| *NCI-MATCH* (Molecular Analysis for Therapy Choice) (NCT02465060) (Flaherty et al., 2020) | Adults (age ≥ 18 years) with advanced solid tumours, lymphomas or myelomas that have progressed after standard treatments or for whom no standard treatment is available | Targeted therapies matched to specific genetic tumour abnormalities. Patients are assigned to different treatment arms based on the genetic alterations found in their tumours | Oncomine Cancer Panel (Lih et al., 2017) based on FFPE-extracted DNA and RNA, IHC (Khoury et al., 2018) | Phase 2, non-randomised, open-label, multicentre clinical trial | ORR | 1,201 | Arms:<br>• *Z1B* (palbociclib in BCa with amp CCND1/2/3: *ORR 0%* (Clark et al., 2023)<br>• *B* (afatinib in pts. with *EGFR2*-activating mutations): *ORR 2.7%* (Bedard et al., 2022)<br>• *F* (crizotinib in *ALK*-rearranged ca): *ORR 50%*<br>• *G* (crizotinib in *ROS1*-rearranged ca): *ORR 25%* (Mansfield et al., 2022)<br>• *I* (taselisib in *PIK3CA*-mutated ca other than BCa and SCC): *ORR 0%* (Krop et al., 2022)<br>• *Z1F* (copanlisib in *PIK3CA*-mut ca): *ORR 16%* (Damodaran et al., 2022)<br>• *Z1A* (binimetinib in *NRAS*-mut ca excluding melanoma): *ORR 2.1%* (Cleary et al., 2021)<br>• *Y* (capivasertib in *AKT1 E17K*-mut ca): *ORR 29%* (Kalinsky et al., 2021)<br>• *H* (dabrafenib and trametinib in *BRAFV600E*-mut ca): *ORR 38%* (Salama et al., 2020)<br>• *W* (AZD4547 in *FGFR* amp/mut/tx ca): *ORR 8%* (Chae et al., 2020)<br>• *R* (trametinib in non-V600 *BRAF* mut ca): *ORR 3%* (Johnson et al., 2020)<br>• *Z1D* (nivolumab in MMRd ca): *ORR 36%* (Azad et al., 2020)<br>• *Q* (Ado-trastuzumab emtansine in HER2-amp ca excluding BCa and GEJ adenoCa): *ORR 5.6%* |
| *TAPUR* (Targeted Agent and Profiling Utilisation Registry) (NCT02693535) (Mangat et al., 2018) | Patients (age ≥ 12 years) with histologically-proven locally advanced or metastatic solid tumours, multiple myeloma or B cell non-Hodgkin lymphoma who are no longer benefiting from standard anticancer treatment or no such treatment is available or indicated | FDA-approved targeted therapies (usually used for other cancer types) matched to the specific genomic alterations in a patient's tumour | NGS | Phase 2, non-randomised, open-label, multicentre clinical trial | ORR | 3,581 (planned) | Two arms closed at stage I because of a lack of responses; 12 arms expanded to stage II<br>• *ALK, ROS1, MET* – crizotinib<br>• *CDKN2A, CDK4, CDK6* – palbociclib or abemaciclib<br>• *CSF1R, PDGFR, VEGFR* – sunitinib<br>• *mTOR, TSC* – temsirolimus<br>• *BRAF V600E/D/K/R* – vemurafenib and cobimetinib<br>• RET, VEGFR1/2/3, KIT, PDGFRβ, RAF-1, BRAF - regorafenib<br>• *BRCA1/2, ATM* – olaparib<br>• *NRG1* Afatinib |

*(Continued)*

| Trial ID/Name | Patient population | Intervention | Precision technologies | Study design | Primary endpoint | N | Results |
|---|---|---|---|---|---|---|---|
| | | | | | | | • *BRCA1/2, PALB2* – talazoparib<br>• *ROS1* fusion – entrectinib<br>• *NTRK* amplification – larotrectinib |
| *I-PREDICT* (Profile Related Evidence Determining Individualised Cancer Therapy) (NCT02534675) (Sicklick et al., 2019) | Adults (age ≥ 18 years) with advanced or metastatic solid tumours that have progressed after standard treatments or for which no standard treatment is available | Personalised targeted therapies and combination treatments based on genomic tumour profiling; the treatment plan designed by a molecular tumour board using DNA sequencing and other molecular analysis techniques to identify actionable genomic alterations | Tissue genomic profiling using NGS (Foundation Medicine; 236–405 genes), PD-L1 IHC, TMB, MSI status, ctDNA | Phase 2, single-arm, open-label, prospective clinical trial | ORR | 149 | *ORR 11.4%*<br>A High Matching Score was an independent predictor of higher DCR (OR 3.6; 95% CI 1.1–11.8; *p = 0.033*) |
| *MyPathway* (NCT02091141) | Adults (age ≥ 18 years) with advanced solid tumours that have progressed after standard treatments or for which no standard treatment is available | Targeted therapies that are matched to specific molecular tumour alterations. The trial investigates the off-label use of targeted therapies which are FDA-approved for other cancer indications | IHC, FISH, NGS, FoundationOne CDx | Tissue-agnostic, non-randomised, phase 2a multiple basket trial | ORR | 357<br>13<br>70<br>37<br>21<br>43 | Arms:<br>• *HER2* (trastuzumab + pertuzumab in *HER2*-altered ca): *ORR 23.3%*<br>• *EGFR* (erlotinib in *EGFR*-mut. ca): *ORR 7.7%*<br>• *BRAF* (vemurafenib ± cobimetinib in *BRAF*-mut. ca): *ORR 24.3%*<br>• *Hh* (vismodegib in *PTCH1/SMO*-mut. ca): *ORR 10.8%*<br>• *ALK* (alectinib in *ALK*-driven ca.): *ORR 30%*<br>• *TMB* (atezolizumab in TMB-high ca): *ORR 39.5%* (Friedman et al., 2022) |
| *LUNG-MAP* (Lung Cancer Master Protocol) (NCT02154490) (Redman et al., 2020) | Adults (age ≥ 18 years) with advanced or metastatic squamous or non-squamous NSCLC who have progressed after first-line standard therapy | Targeted therapies and immunotherapies matched to specific tumour molecular alterations. The trial investigates the use of these therapies in patients with advanced NSCLC who have progressed after first-line standard therapy | NGS (FoundationOne) IHC | Phase 2/3, randomised, open-label, multicentre clinical trial | ORR; OS in phase 3 substudies | 1864 | Substudies:<br>• S1400A (durvalumab vs. docetaxel)<br>• S1400B (taselisib vs. docetaxel in *PI3KCA*-mut. ca)<br>• S1400C (palbociclib vs. docetaxel in *CDK4/6, CCND1/2/3*-positive ca)<br>• S1400D (AZD4547 vs. docetaxel in pts. positive for FGFR1/2/3)<br>• S1400E (rilotumumab + erlotinib vs. erlotinib in *HGF/c-MET*-pos. ca)<br>• S1400F (durvalumab + tremelimumab)<br>• S1400G (talazoparib in HHR-deficient ca)<br>• S1400I (nivolumab + ipilimumab vs. nivolu-mab) |
| *BATTLE* (Biomarker-integrated Approaches of Targeted Therapy for Lung Cancer Elimination) (E. | Adults (age ≥ 18 years) with advanced NSCLC who have progressed after first-line platinum-based chemotherapy | Erlotinib (an EGFR inhibitor), vandetanib (a VEGFR inhibitor), erlotinib plus bexarotene (a retinoid X receptor agonist) and sorafenib (a multikinase inhibitor). Patients assigned to one of these treatments based on the molecular | FISH, IHC, Sanger sequencing | Phase 2, adaptive, randomised, open-label clinical trial | 8-week DCR | 255 | The 8-week 46% DCR for the entire study population, with the following rates for each treatment arm:<br>• erlotinib: *43%*<br>• vandetanib: *39%*<br>• erlotinib plus bexarotene: *50%*<br>• sorafenib: *53%* |

**Table 1.** (*Continued*)

| Trial ID/Name | Patient population | Intervention | Precision technologies | Study design | Primary endpoint | N | Results |
|---|---|---|---|---|---|---|---|
| S. Kim et al., 2011) | | tumour profile, assessed using biomarkers, such as EGFR, KRAS, VEGF and cyclin D1 | | | | | |
| *SHIVA* (NCT01771458) (Le Tourneau et al., 2015) | Adults (age ≥ 18 years) with advanced solid tumours that have progressed after standard treatments or for which no standard treatment is available | Patients randomised into two groups: the experimental group receiving molecularly targeted agents based on tumour molecular profiling, and the control group receiving treatment according to the physician's choice | Targeted NGS, Cytoscan copy number analysis | Phase 2, open-label, randomised, controlled clinical trial | PFS | 293 | Median PFSl 2.3 months (95% CI 1.7–3.8) in the experimental group vs. 2.0 months (1.8–2.1) in the control group (hazard ratio 0·88, 95% CI 0.65–1.19, *p* = 0.41) |
| *WINTHER* (Worldwide Innovative Networking in Personalised Cancer Medicine) (NCT01856296) | Adults (age ≥ 18 years) with advanced solid tumours that have progressed after standard treatments or for which no standard treatment is available | Personalised targeted therapies and chemotherapies that are matched to specific genomic alterations or gene expression patterns in the patient's tumour | Fresh biopsy: DNA + RNA NGS testing | Phase 2, non-randomised, open-label, multicentre clinical trial | PFSr | 107 | The trial did not meet its primary endpoint, as the PFS ratio of ≥1.5 was observed in only 22% of patients in Arm A (DNA-seq-based drug matching) and 26% in Arm B (RNA-seq-based drug matching) |
| *GBM AGILE* (Glioblastoma Adaptive Global Innovative Learning Environment) (NCT03970447) (Alexander et al., 2018) | Adults (age ≥ 18 years) with newly diagnosed or recurrent glioblastoma | Targeted therapies and immunotherapies, which are compared to standard treatment options. The specific agents tested in the trial may change over time as new treatments become available or others are dropped based on their performance in the study | NR | Phase 2/3, adaptive, randomised, open-label, multicentre clinical trial | PFS/OS | NR | NR |
| *FOCUS4* (Brown et al., 2022) | Adults (age ≥ 18 years) with advanced colorectal cancer who have completed 16 weeks of first-line chemotherapy | Patients are first treated with standard chemotherapy and then, based on their molecular subtyping, are randomised into different treatment arms; the trial tests these targeted therapies against a control group receiving standard treatment or a placebo, and the specific agents tested in the trial may change over time as new treatments become available or others are dropped based on their performance in the study | Pyrosequencing, NGS, IHC (Richman et al., 2022) | Phase 2/3, randomised, open-label, multicentre clinical trial | PFS/OS | 361 | FOCUS4-D (sapitinib in *BRAF-PIK3CA-RAS* wt ca) closed FOCUS4-B (aspirin in *PIK3CA*-mut. ca) closed FOCUS4-C (adavosertib in *RAS* + *TP53* double mutant) (Seligmann et al., 2021) FOCUS4-N (nonstratified). Median PFS in the capecitabine arm 3.9 months (95% CI 3.7–4.4) and 1.9 months (95% CI 1.8–2.1) in the AM arm/Unadjusted and adjusted HRs 0.44 (95% CI 0.33–0.57), *p* < 0.0001 and 0.40 (95% CI 0.21–0.75), *p* < 0.0001, respectively |

(*Continued*)

**Table 1.** (*Continued*)

| Trial ID/Name | Patient population | Intervention | Precision technologies | Study design | Primary endpoint | N | Results |
|---|---|---|---|---|---|---|---|
| *NCI-COG Paediatric MATCH* NCT03155620 | Children and adolescents (aged 1–21 years old) with recurrent, refractory or progressive solid tumours, lymphomas and histiocytic disorders | Molecularly targeted therapies matched to specific tumour genetic alterations | DNA and RNA sequencing, IHC (Parsons et al., 2022) | Phase 2, open-label, multicentre clinical trial | ORR | 2,316 (planned) 20 20 20 | Subprotocols: <br> • A (larotrectinib in ca with *NTRK* fusions) <br> • B (erdafitinib in ca with *FGFR1/2/3/4* mutation) <br> • C (patients with an *EZH2, SMARCB1 or SMARCA4* mutation receive tazemetostat) ORR: 1 response (Chi et al., 2022) <br> • D (samotolisib in patients with *TSC1, TSC2* or *PI3K/mTOR* mutations) <br> • E (selumetnib in *MAPK*-mut. ca) ORR 0% (Eckstein et al., 2022) <br> • F (ensartinib in ca with *ALK* or *ROS1* alterations) <br> • G (vemurafenib in *BRAF V600E*-pos. ca) <br> • H (olaparib in pts. with *ATM, BRCA1, BRCA2, RAD51C* or *RAD51D* mut.) <br> • I (palbociclib in pts. with *Rb*-positive ca) <br> • J (ulixertinib in pts. with *MAPK*-mut. ca) (Vo et al., 2022): ORR 0% <br> • K (ivosidenib in ca with *IDH1* mutations) <br> • M (tipifarnib in ca with *HRAS* alterations) <br> • N (selpercatinib in ca with *RET* alterations) |
| *SAFIR02-BREAST* (NCT02299999) (Mosele et al., 2020; Andre et al., 2022) | Adult women (age ≥ 18 years) with metastatic breast cancer who have progressed after standard treatments or for whom no standard treatment is available | Molecularly targeted therapies matched to specific genomic alterations in the patient's tumour | NGS, CGH array | Phase 2, open-label, multicentre clinical trial | PFS | 436 157 50 40 20 17 17 7 3 3 131 148 | Arms: <br> Arm A1 (targeted arm): Patients in this arm received targeted maintenance therapy guided by genomic analysis. The therapy used eight targeted drugs. ORR: 26/157 <br> • Capivasertib in *PI3K/AKT*-altered ca <br> • Olaparib in *BRCA/DDR*-altered ca <br> • Alpelisib in PI3KCa-mut ca <br> • Selumetinib in *MAPK*-altered ca <br> • AZD4547 in *FGFR*-mut. ca <br> • Vistusertib in mTOR-altered ca <br> • Sapitinib in *HER2/3*-altered ca <br> • Vandetanib <br> Arm A2 (immunotherapy arm): Patients in this arm received durvalumab <br> Arm B (standard maintenance chemotherapy): Patients in this arm received standard maintenance chemotherapy. ORR 9/81 (Bachelot et al., 2021) <br> After a median follow-up of 21.4 months (90% CI: 17.9–27.6), patients with ESCAT I/II showed a significantly longer PFS in the targeted therapy arm than in the control arm, with a median PFS of 9.1 months (90% CI 7.1–9.8) and 2.8 months (90% CI 2.1–4.8), respectively (adjusted HR = 0.41, 90% CI 0.27–0.61; $p < 0.001$) |

(*Continued*)

**Table 1.** (*Continued*)

| Trial ID/Name | Patient population | Intervention | Precision technologies | Study design | Primary endpoint | N | Results |
|---|---|---|---|---|---|---|---|
| *IMPACT* (Integrated Molecular Profiling in Advanced Cancers Trial) NCT01505400 (Stockley et al., 2016) *COMPACT* (Community Molecular Profiling in Advanced Cancers Trial) | Adults (age ≥ 18 years) with histological confirmation of advanced breast, non-small cell lung, colorectal, genitourinary, pancreatobiliary gastrointestinal, upper aerodigestive tract, gynaecological, melanoma, unknown primary, and rare carcinomas who are candidates for systemic therapy, as well as patients who are phase 1 trial candidates | Two parallel trials that aim to identify molecular alterations in patients' tumours and match them to targeted therapies, with IMPACT being conducted at academic centres and COMPACT in community settings | Targeted NGS panel and MALDI-TOF-based multiplex genotyping panel | Retrospective cohort study | NR | 1893 | ORR higher in patients treated on genotype-matched (19%) than in genotype-unmatched trials (9%; $p = 0.026$). In multivariate analysis, trial matching according to genotype ($p = 0.021$) and female gender ($p = 0.034$) were the only statistically significant factors associated with response. Genotype-matched trial patients were more likely to achieve the best response of any shrinkage in the sum of their target lesions (62%) compared with genotype-unmatched trial patients (32%; $p < 0.001$) |
| *MOSCATO 01* (Massard et al., 2017) | Adults (age ≥ 18 years) with advanced solid tumours that have progressed after standard treatments or for whom no standard treatment is available | Molecularly targeted therapies that are matched to specific genomic alterations in the patient's tumour | aCGH, WES, RNA-seq | Phase 2, non-randomised, open-label, multicentre clinical trial | PFSr | 843 | Molecular profiling and matching patients to targeted therapies led to an improvement in the ORR (11% vs. 5%) and PFS. Progression-Free Survival Ratio (PFS2/PFS1) >1.3 in 33% of patients. Following targeted therapy, of the evaluable patients, two had CR and 20 PR |
| *INFORM* (individualised Therapy FOr Relapsed Malignancies in Childhood) (van Tilburg et al., 2021) | Children and adolescents (age ≤ 21) with relapsed or refractory malignancies, including solid tumours, lymphomas and central nervous system tumours | Molecularly targeted therapies and immunotherapies matched to specific genomic alterations and immunological tumour features | WES, lcWGS, RNA sequencing, RNA-based gene expression array and DNA-methylation | Prospective, noninterventional, multicentre, multinational and feasibility registry | NR | 519 | No significant differences in PFS and OS in all patients who did and did not receive a matched targeted drug |
| *MINDACT* (Microarray in Node-Negative and 1 to 3 Positive Lymph Node Disease May Avoid Chemotherapy) | Women (age ≥ 18 years) with histologically proven, operable, invasive, early-stage breast cancer who have node-negative or 1 to 3 positive lymph nodes | Use of a 70-gene signature (MammaPrint) to determine the likelihood of distant recurrence in women with early-stage breast cancer. The trial compared the outcomes of patients assigned to adjuvant chemotherapy based on the traditional clinical-pathological assessment and the MammaPrint assay | MammaPrint | Phase 3, randomised, controlled, multicentre clinical trial | 5-year DMFS rate | 6,693 | The primary endpoint was met; the inferior margin of 92.5% for DMFS at 60 months in the targeted subjects exceeded the 92% pre-specified threshold |
| *ALCHEMIST* (Adjuvant Lung Cancer Enrichment Marker Identification | Adults (age ≥ 18 years) with surgically resected stage IB, II or IIIA NSCLC | Three separate subtrials, each using a specific targeted therapy based on the presence of particular tumour genomic alterations. Investigated targeted | EGFR sequencing, ALK FISH, PD-L1 IHC | Three integrated, phase 3, randomised, double-blind, placebo-controlled clinical trials | OS | 4,405 | NR |

(*Continued*)

**Table 1.** (*Continued*)

| Trial ID/Name | Patient population | Intervention | Precision technologies | Study design | Primary endpoint | N | Results |
|---|---|---|---|---|---|---|---|
| and Sequencing Trial) | | therapies: erlotinib (for patients with EGFR mutations), crizotinib (for patients with ALK rearrangements) and nivolumab (for patients with high PD-L1 expression) | | | | | |
| *DRUP* (The Drug Rediscovery Protocol) (NCT02925234) (Hoes et al., 2022) | Adults (age > 18 years) with a histologically-proven locally advanced or metastatic solid tumour, multiple myeloma, or B cell non-Hodgkin lymphoma who are no longer benefitting from standard anti-cancer treatment or for whom no such treatment is available or indicated | The molecular profiling test results are used to determine appropriate drugs from those available in the protocol. The choice of the drug is supported by a list of potential profiles, a molecular tumour board, a knowledge library, and study coordinators for review and approval of the match | Fresh biopsy: WGS; off-label use | Phase II, prospective, non-randomised basket trial | ORR | 1,550 (planned) | NR |
| *TARGET* (Tumour Characterisation to Guide Experimental Targeted Therapy) (NCT04723316) (Rothwell et al., 2019) | Patients aged 16 years or over with confirmed diagnosis of advanced solid cancer | The primary aim of TARGET National is to establish a national framework to offer molecular profiling of circulating tumour DNA and/or tumour tissue (optional) to patients with advanced solid cancers | ctDNA testing | Prospective observational trial | Number of patients matched to a trial of an experimental therapeutic agent based on molecular findings from ctDNA or tumour | 6,000 (planned) | NR |
| *TAPISTRY* (Tumour-Agnostic Precision Immuno-Oncology and Somatic Targeting Rational for You) (NCT04589845) | Patients with confirmed diagnosis of advanced and unresectable or metastatic solid malignancy | Study evaluating the efficacy and safety of targeted therapy or immunotherapy, as single agents or in combination, in patients with unresectable, locally advanced or metastatic solid tumours divided into cohorts based on biomarkers | NGS, FoundationOne CDx / FoundationOne Liquid CDx-based assays | Phase II, global, multicenter, open-label, multi-cohort trial | ORR | 770 (planned) | NR |

ALK, anaplastic lymphoma kinase; CBR, clinical benefit rate; CI, confidence interval; ctDNA, circulating tumour DNA; DCR, disease control rate; DMFS, distant metastasis-free survival; DNA, deoxyribonucleic acid; EGFR, epidermal growth factor receptor; FDA, Food and Drug Administration; FFPE, formalin-fixed, paraffin-embedded; ICD − O, International Classification of Diseases for Oncology; IHC, immunohistochemistry; KRAS, Kirsten Rat Sarcoma Viral oncogene homologue; MSI, microsatellite instability; MTB, molecular tumour board; NGS, next-generation sequencing; NR, not reported; NSCLC, non-small cell lung cancer; OR, odds ratio; ORR, objective response rate; OS, overall survival; PD-L1, programmed death-ligand 1; PFS, progression-free survival; PFSr, PFS interval associated with molecularly informed therapy (PFS2) divided by the PFS interval associated with the last prior systemic therapy (PFS1); RNA, ribonucleic acid; RNA-seq, RNA sequencing; TMB, tumour mutational burden; VEGF, vascular endothelial growth factor; VEGFR, vascular endothelial growth factor receptor; WES, whole exome sequencing; WGS, whole genome sequencing.

targeted and personalised radiotherapy regimens tailored to the specific characteristics of individual patients and their tumours. Cancer heterogeneity poses a significant challenge for radiotherapy, as it can cause variable tumour responses and the emergence of radioresistant cell populations. Precision radiotherapy, by considering the comprehensive molecular and genetic tumour makeup, may overcome these challenges and allow for the development of tailored treatment plans. By integrating genomic data and other biomarkers, precision radiotherapy has the potential to maximise tumour control while minimising toxicity to surrounding healthy tissues.

### Tumour response

Technological advancements in radiotherapy have increased the potential of physical radiation tailoring to personalise treatment. However, the optimisation process typically focuses on dose conformality, ignoring biological factors and assuming that all tumours react similarly to radiation (Price et al., 2023). Unlike medical oncology, where genomic signatures have become part of routine practice (e.g., MammaPrint tests, Oncotype DX or PAM50), their use in radiotherapy has been limited (Parker et al., 2023). Meanwhile, radiation impacts several molecular pathways, such as DNA damage, hypoxia or proliferation (Reisz et al., 2014; Wang et al., 2018; Huang and Zhou, 2020).

Several somatic mutations have already been established as conferring radioresistance. Numerous studies, including breast (Jameel et al., 2004), colorectal (Munro et al., 2005) and head and neck cancers (Hutchinson et al., 2020), gliomas (Werbrouck et al., 2019) and sarcomas (Casey et al., 2021) have shown that *TP53* mutations might impair radiotherapy response. Other notable examples are *KEAP1* and *NFE2L2/NRF2* mutations in NSCLC and head and neck cancers (Binkley et al., 2020; Guan et al., 2023). In addition, the coexistence of *KRAS* and *SMAD4* mutations is an indicator of radioresistance in cervical cancer (Oike et al., 2021).

To date, there has been scarce data on molecular predictive signatures in radiotherapy. These examples comprise the PORTOS classifier encompassing 24 genes to predict the efficacy of postoperative radiotherapy in prostate cancer (Zhao et al., 2016) and the Adjuvant Radiotherapy Intensification Classifier (ARTIC) and POLAR classifiers, which incorporate 27 and 16 genes, respectively, to predict outcomes of postoperative radiotherapy in breast cancer (Sjöström et al., 2019, 2023).

Unlike PORTOS or POLAR, which relate to specific cancers and clinical situations, a radiosensitivity index (RSI) has also been proposed as a pan-cancer and specific marker of cellular radiosensitivity. This index is based on the expression of 10 genes (*AR, c-JUN, STAT1, PKC, Rel A, cABL, SUMO1, CDK1, HDAC1* and *IRF1*) related to DNA damage response, cell cycle, apoptosis and proliferation (Eschrich et al., 2009). Based on RSI, a quantitative metric for the biological effect of RT, the genomic-adjusted radiation dose (GARD) has been developed. GARD was initially validated in patients with breast cancer, lung cancer, pancreatic cancer and glioblastoma (Scott et al., 2017). This signature was further tested in a pooled, retrospective, pan-cancer cohort and reported as a continuous variable associated with time to first recurrence and overall survival (Scott et al., 2021). Recently, GARD has been employed in a provocative in silico analysis to explain the unexpected results of the seminal RTOG 0617 trial (unsuccessful radiotherapy dose escalation in locally advanced NSCLC) (Scott et al., 2021). The authors assumed that this model allows for deriving an optimal radiation dose in each patient. Another study employing

prospectively collected tissues showed that low RSI values (denoting higher radiosensitivity) are associated with increased immune infiltration and activation (Grass et al., 2022). Recently, based on the reanalysis of the publicly available datasets – Merged Microarray-Acquired Dataset (Bin Lim et al., 2019) and the Cancer Genome Atlas (Weinstein et al., 2013) – RSI was shown to be associated with immune-related features and predicted response to PD-1 blockade (Dai et al., 2021). However, a recent analysis showed that RSI is not associated with survival and should not be used for radiation dose adjustments (Mistry, 2023). It was also suggested that the RSI of tumour clones remaining after RT, instead of the initial tumour population, should be evaluated to better predict the RT outcome (Alfonso and Berk, 2019).

Incorporating genomic signatures in radiotherapy decision-making has shown significant advancement through recent research, such as the GARD-based trial, to optimise radiotherapy for triple-negative breast cancer (NCT05528133). The European Organisation for Research and Treatment of Cancer has appraised the evidence from RSI/GARD studies as a priority for phase 3 clinical trials in radiotherapy (Thomas et al., 2020). However, the clinical utility of these approaches warrants an evaluation that integrates molecular data into prospective clinical trials and routine clinical practice (Table 2).

### Radiotherapy tolerance

The impact of genetic heterogeneity on normal tissue toxicity following radiotherapy is a significant concern in cancer treatment. Individual genetic variations can influence the severity of radiation-induced side effects (Barnett et al., 2009). Normal tissue complications can range from mild to severe and may include skin reactions, inflammation, fibrosis and organ dysfunction (Bentzen, 2006). However, except for several radiosensitivity syndromes related to biallelic pathogenic mutations in DNA repair genes and deleterious heterozygous ATM mutations in young patients, no genomics-guided radiotherapy is currently used (Bergom et al., 2019).

Since the beginning of the twenty-first century, more than 100 articles analysing the impact of DNA sequence changes on the frequency and severity of radiation-induced complications have been published (Andreassen et al., 2016). Most of these studies have addressed SNVs, which typically affect the genes responsible for processes such as DNA break or inflammation. However, these studies were usually small (median of approximately 150 patients), hence lowering the statistical power for comparisons (Andreassen et al., 2016).

To reduce the bias associated with the publication of numerous low-quality studies, the Radiogenomics Consortium (https://epi.grants.cancer.gov/radiogenomics/) was created in 2009 (West et al., 2010). This initiative allowed for assembling adequate groups of patients with diverse clinical characteristics and validating presumed associations of SNVs with radiation toxicity. However, the results of the prospective study published in 2012 were a huge disappointment because none of the reported relationships (98 SNVs in 46 genes) were confirmed (Barnett et al., 2012). However, this experience prompted the development of research employing large-scale techniques such as genome-wide association studies (GWAS). As a result, potentially interesting SNV associations with radiation reactions were found, such as variants at the locus of the *TANC1* gene that was found to be encoding a protein responsible for muscle cell regeneration (Fachal et al., 2014). The strength of these associations is much higher, with odds ratios of 1.3–1.5, compared with 1.1–1.2 observed in typical GWAS studies

**Table 2.** Summary of selected clinical studies investigating radiosensitivity-predicting genomic signatures

| Study | Cancer type | Sample size | Main findings |
|---|---|---|---|
| Zhao et al., 2016 | Prostate cancer | 526 patients (196 and 330 in training and validation cohorts, respectively) | 24-gene predictor of response to postoperative RT. High PORTOS score predicted a lower incidence of distant metastases in both training (HR 0.12; 95%CI: 0.03–0.41; $p < 0.0001$) and validation (HR 0.15; 95% CI 0.04–0.60; $p = 0.002$) cohorts |
| Tang et al., 2017 | Sarcomas | 253 patients from The Cancer Genome Atlas | 26-gene radiosensitivity signature. Predicted radiosensitive patients had better overall survival than predicted nonradiosensitive patients (HR 0.07, $p < 0.001$) |
| Cui et al., 2018 | Breast cancer | 948 and 1,439 patients in the training and validation cohorts, respectively (METABRIC) | 34-gene radiosensitivity signature. Patients administered RT had better disease-specific survival than those who did not in the radiation-sensitive group (HR 0.68, $p = 0.059$); a reverse effect was observed in the radiation-resistant group (HR 1.53, $p = 0.059$)<br>4-gene immune signature predictive of RT benefit. Patients who were administered RT had significantly better disease-specific survival in the immune-effective group (HR 0.46, $p = 0.0076$), with no difference in disease-specific survival in the immune-defective group (HR 1.27, $p = 0.16$) |
| Sjöström et al., 2019 | Breast cancer | 748 patients from the SweBCG91-RT trial | Adjuvant Radiotherapy Intensification Classifier (ARTIC) comprising 27 genes and patient age was prognostic for locoregional recurrence in breast cancer patients administered RT (HR 3.4; 95% CI: 2.0 to 5.9; $p < 0.001$) and was predictive of RT benefit ($p = 0.005$). Patients with low ARTIC scores had a larger benefit from RT (HR 0.33; 95% CI: 0.21 to 0.52, $p < 0.001$) than those with high ARTIC scores (HR 0.73; 95% CI: 0.44 to 1.2, $p = 0.23$) |
| Kim et al., 2020 | HPV-negative head and neck squamous cell carcinomas | 203 patients from The Cancer Genome Atlas (TCGA) cohort | 41-gene radiation sensitivity signature (RSS). RSS predicted reduced 5-year recurrence-fee survival in the radioresistant group versus the radiosensitive group (57.8% vs. 80.1%; $p = 0.035$) |
| Scott et al., 2021 | Various types (breast, head and neck, NSCLC, pancreatic, endometrial, melanoma and glioma) | 1,615 patients, of whom 1,298 (982 and 316 with and without RT, respectively) assessed for time to first recurrence and 677 (424 and 253 with and without RT, respectively) for overall survival | Genomic-adjusted radiation dose (GARD) was associated with time to first recurrence (HR 0.98, 95% CI 0.97–0.99; $p = 0.0017$) and overall survival (HR 0.97, 0.95–0.99; $p = 0.0007$). The effect on overall survival was dependent on radiotherapy use ($p = 0.011$) |
| Feng et al., 2021 | Prostate cancer | 486 of 760 patients randomised in NRG/RTOG 9601 trial | 22-gene genomic classifier Decipher (Decipher Biosciences Inc) was associated with distant metastases (HR 1.17; 95%CI: 1.05–1.32; $p = 0.006$), prostate cancer–specific mortality (HR 1.39; 95%CI: 1.20–1.63; $p < 0.001$) and overall survival (HR 1.17; 95%CI: 1.06–1.29; $p = 0.002$) |
| Dal Pra et al., 2022 | Prostate cancer | 226 of 350 patients randomised in Swiss Group for Clinical Cancer Research (SAKK) 09/10 trial | 22-gene genomic classifier Decipher (Decipher Biosciences Inc) was associated with biochemical progression (HR 2.26; 95%CI: 1.42–3.60; $p < 0.001$), clinical progression (HR 2.29, 95%CI: 1.32–3.98; $p = 0.003$) and use of hormone therapy (sHR 2.99, 95% CI 1.55–5.76; $p = 0.001$). Patients with high and low Decipher scores had 45% and 71% 5-year freedom from biochemical progression, respectively |
| Wu et al., 2022 | Gliomas | 1,395 from Chinese Glioma Genome Atlas and Cancer Genome Atlas | 12-gene radiosensitivity predictive index (PI12) had better predictive capacity than the traditional WHO classification system (concordance index: 0.842 vs. 0.787; $p \leq 2.2 \times 10^{-16}$) |
| Sjöström et al., 2023 | Breast cancer | 729 patients from the SweBCG91-RT trial and Princess Margaret Hospital | A 16-gene signature named Profile for the Omission of Local Adjuvant Radiation (POLAR). POLAR low-risk patients did not benefit from adjuvant RT (HR 1.1; 95% CI 0.39–3.40; $p = 0.81$; HR 1.5; 95% CI 0.14–16, $p = 0.74$). POLAR high-risk patients had a significantly lower risk of locoregional recurrence with RT (HR 0.43; 95% CI 0.24–0.78; $p = 0.006$; HR 0.25; 95%CI 0.07–0.92; $p = 0.038$) |
| Spratt et al., 2023 | Prostate cancer | 215 patients from NRG Oncology/RTOG 0126 | 22-gene genomic classifier Decipher (Decipher Biosciences Inc) was independently prognostic for disease progression (sHR 1.12; 95%CI 1.00–1.26, $p = 0.04$), biochemical failure (sHR 1.22; 95%CI 1.10–1.37, $p < 0.001$), distant metastasis (sHR 1.28; 95%CI 1.06–1.55, $p = 0.01$) and prostate cancer-specific mortality (sHR 1.45; 95%CI 1.20–1.76, $p < 0.001$) |

95%CI, 95% confidence interval; HR, hazard ratio; sHR, subdistribution hazard ratio.

(Zhong and Prentice, 2010). The Radiogenomics Consortium remains active, and a significant increase in sample size has led to the discovery of several potentially relevant relationships. An analysis of breast and prostate cancer patients from 17 cohorts indicated that the *ATM* rs1801516 SNP is associated with an increased risk of radiation toxicity (Andreassen et al., 2016). A recent study has revealed a strong association between radiation-induced mucositis and the rs1131769*C locus in the *STING1* gene on chromosome 5 (Schack et al., 2022).

A definitive answer to the SNVs' role in healthy tissues' response to radiation may come from the international, multicentre REQUITE project (www.requite.eu) funded by the European Union through its 7th Framework Programme (West et al., 2014). The project, performed in collaboration with the

Radiogenomics Consortium, aimed at predicting and reducing the risk of long-term side effects of radiotherapy and completed patient recruitment (Seibold et al., 2019). REQUITE reported that polygenic risk scores (PRS) may be clinically useful and that incorporation of SNP-SNP interactions improves patient classification and prediction of radiotherapy-related toxicity (Franco et al., 2021). This project significantly advanced the collaborations among stakeholders, including healthcare professionals, researchers and industry partners, highlighting the importance of personalised radiotherapy. Other collaborative genetic association studies at both the national and global levels include Gene-PARE (Ho et al., 2006), RadGenomics (Iwakawa et al., 2002) and RAPPER (Burnet et al., 2013). Understanding the impact of radiogenomic heterogeneity on normal tissue radiation toxicity is essential for developing more effective and safe personalised strategies.

In summary, for a long time, radiotherapy optimisation was focused on dosage conformity rather than biological factors. It is critical to factor in tumour heterogeneity when considering radiotherapy outcomes; hence, developing and verifying molecular signatures such as PORTOS, ARTIC and POLAR, together with pan-cancer RSI, constitute the base for resolving this predicament.

## Other factors

### Circadian heterogeneity

The efficacy of anticancer treatment may also be affected by circadian rhythm, a biological phenomenon displaying endogenous, untrainable 24-h oscillation (Lee, 2021). The circadian clock regulates several key processes in the human body, including metabolism and cell division, with 40% of the transcriptome under circadian control in at least some tissues (Ruben et al., 2018). Recently, a comprehensive analysis of clock genes across different human cancers was performed using primary solid tumour data from The Cancer Genome Atlas (Ye et al., 2018). Based on the available evidence, the International Agency for Research on Cancer classified shift work that involves circadian disruption as potentially carcinogenic to humans (IARC Monographs Vol 124 group, 2019).

Chronotherapy involves administering treatment at specific times of day to optimise its effectiveness and minimise side effects (Zhou et al., 2021). This approach has been tested in several clinical trials with conflicting results: some showed improved efficacy and reduced toxicity (Lévi et al., 1997; Giacchetti et al., 2006), whereas others did not demonstrate significant differences (Garufi et al., 2006; Qvortrup et al., 2010). Some data indicate that chronomodulation might be relevant in the context of immunotherapy. For instance, a recent provocative study reported inferior overall survival in patients who received more than 20% of immunotherapy infusions after 4:30 PM (Qian et al., 2021). However, these observations warrant verification in prospective randomised clinical trials. Data pertinent to radiotherapy comes from the REQUITE project, which disclosed novel serendipitous associations, for example, the interaction between time, circadian rhythm-related genes (*CLOCK*, *PER3* and *RASD1*) and late radiation toxicity in breast cancer patients (Webb et al., 2022).

### Microbiome heterogeneity

The human microbiome, comprising trillions of diverse microbial organisms, plays a significant role in modulating health and disease states, including cancer (Hou et al., 2022). Recent research indicates that microbiome heterogeneity can greatly influence response to anticancer treatment via drug-microbiota interactions.

Bacteria-derived enzymes target chemical compounds, including drugs used in systemic treatment. For example, approximately 40% of patients treated with irinotecan experience severe mucositis, sometimes leading to treatment cessation. Irinotecan is converted into its active form, SN38, which is later reverted back to an inactive form, SN38G, in the liver. Bacterial β-glucuronidases can then convert SN38G in the gastrointestinal tract back to its toxic form (Wallace et al., 2010). Additionally, *Bifidobacterium longum, Collinsella aerofaciens and Enterococcus faecium* abundancy in stool was found to be associated with increased response to anti–PD-1 treatment in patients with melanoma (Matson et al., 2018).

Additionally, the gut microbiome generates numerous metabolites, with short-chain fatty acids (SCFAs) being among the most prevalent and crucial. Significantly, SCFAs act as secondary messengers that facilitate signal transmission and influence the onset and progression of various diseases. Radiotherapy can modify the populations of bacteria that produce SCFAs, leading to changes in SCFA levels, which are linked to several conditions, including radiation-induced intestinal injury (Li et al., 2021).

The microbiome studies shape oncologic outcomes and are now being leveraged for the development of novel personalised therapeutic approaches in anticancer treatment. However, this topic exceeds the scope of this paper and has been addressed elsewhere (Chrysostomou et al., 2023; Yi et al., 2023).

## Conclusions

Precision medicine has made remarkable progress in oncology by promising to administer therapy to "the right patient at the right time" (Abrahams, 2008). This review has discussed the impact of cancer heterogeneity as a major challenge facing precision oncology development. Apart from affecting treatment outcomes, heterogeneity can also be employed in the context of prevention and early detection. The results of the first large-scale observational cohort study evaluating methylation-based multicancer early detection diagnostic test (SIMPLIFY) have demonstrated the feasibility of this approach (Rebbeck et al., 2018; Nicholson et al., 2023; Tie, 2023).

Genomic makeup has been shown to impact the effectiveness and toxicity of systemic treatments and radiotherapy. Thus, genomic testing can identify pathogenic gene variants and polymorphisms affecting drug metabolism or mechanism of action, thus increasing the risk of treatment failure or toxicity. Spatial and temporal tumoural heterogeneity is a complex phenomenon linked to resistance to therapy, disease progression and adverse prognosis. There are substantial genetic and molecular differences across various tumour regions and between primary and metastatic foci. A better understanding of this phenomenon allowed for the development of novel strategies, for example, targeting with systemic therapies or radiation-specific tumour regions or populations of cancer cells.

Implementing advanced technologies, such as NGS, liquid biopsies and imaging modalities, has fostered precision oncology, accounting for both genomic and tumoural heterogeneity. NGS is routinely used to examine mutations in, for example, *EGFR*, *BRAF* and *ALK*, which are molecular targets for modern therapies. Liquid biopsies, which involve analysing circulating tumour cells or circulating tumour DNA, offer a noninvasive way to identify genetic alterations and monitor tumour progression. Several clinical trials,

including the ongoing NCI-MATCH and MASTER trials and the previously completed MyPathway and MPACT trials, have been designed to identify genetic mutations associated with specific targeted therapies and develop novel treatment strategies for overcoming therapy resistance. These trials are expected to prompt further development of precision oncology.

As precision oncology continues to evolve, the future holds great promise for overcoming current challenges. Advanced tools should be more accessible and affordable. There is a need for routine, more comprehensive genomic profiling of different regions of primary and metastatic tumours to fully understand cancer heterogeneity. In addition, integrating machine learning algorithms and artificial intelligence would allow better identification of new therapeutic targets and the development of even more personalised treatment strategies. An exciting area of future research in precision oncology is the use of combination therapies that simultaneously target multiple pathways and molecular targets; this approach has the potential to overcome heterogeneity-led resistance to single-agent targeted therapies. Another area for improvement is integrating precision oncology into clinical practice and expanding access to new technologies for community oncologists and patients. This will require the development of user-friendly platforms and tools that are easily integrable into clinical workflows.

Overall, precision oncology holds great promise for improving cancer treatment efficacy by enabling personalised treatment strategies based on unique cancer and patient characteristics. Although challenges remain to be addressed, ongoing research and emerging developments create real hope for breath-taking therapeutic approaches and improved patient outcomes.

**Open peer review.** To view the open peer review materials for this article, please visit http://doi.org/10.1017/pcm.2023.23.

**Author contribution.** J.J. and B.T. developed the paper concept and outline. B.T., F.G., M.Br. and M.Bi. collected and assembled the data. J.J. provided administrative support and oversight and revised the text. F.G. prepared the figures. B.T., F.G., M.Br., M.Bi. and J.J. wrote and approved the manuscript. B.T. and F.G. contributed equally to this work.

**Financial support.** F.G. is supported by Polish National Science Centre Grants PRELUDIUM 2018/31/N/NZ5/03214, ETIUDA 2020/36/T/NZ5/00610, and Polish Stem Cells Bank grant ExCELLent.

**Competing interest.** J.J. declares advisory roles for BMS, Roche and MSD, lectures for Roche (not compensated), Pfizer, Novartis and MSD and travel support from Takeda. Other authors have no conflicts of interest to declare.

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
