## [Reviewer Report]

Dear Authors,

This is a well-written summary of current concepts in precision oncology. Figures and tables are well done, even though a better described, structured and reproducible search of the literature that led to tables 1 and 2 would be appreciated.

However, there are a few suggestions, that I would like to make in order to improve the manuscript.

Major comments:

- Structure of the article. Your overall topic is diversity in precision oncology. The graphical abstract lists intratumoral, intersite and interpatient heterogeneity. However, the manuscript is not structured in the same way. The introduction of germline (i.e. interpatient?), circadian heterogeneity followed by a section on radiotherapy makes it a little difficult to follow the flow of the manuscript. Furthermore the selection of circadian heterogeneity over e.g.. microbiome diversity seems somewhat random.

Maybe it could help to introduce the concepts of predictive, prognostic, pharmacogenomic and predisposing biomarkers as an overall concept to help navigate diversity?

Minor Comments

- Figure 1 showing highlights of clinical precision medicine (“the” should be omitted, since the selection is somewhat subjective)

- On page 4 you first introduce the concept of selecting chemotherapy according to predictive biomarkers (e.g. BRCA for platinum or MGMT temozolomide) and then state that “Whereas cytotoxic chemotherapy is an important treatment option for many cancer types, it is increasingly recognised that a one-size-fits-all approach may not be optimal for patients with genetically diverse tumours”. It might be better to first introduce the concept of standard systemic therapy without the use of a predictive biomarker followed by the introduction of specific predictive biomarkers (which can also work for cytotoxic chemotherapy)

- BRAF should be carefully selected as an example of tumor-specific activity following the recent pan-cancer approval of dabrafenib/trametinib for BRAF V600E tumors by the FDA

- page 5 “but that are not effective in gliomas bearing these mutations” requires a citation

- page 5 “EGFR-mutant NSCLCs are less sensitive to ICIs than wild-type EGFR” requires a citation

---

## [Editor Report]

This is a concise and timely review of some key aspects of Precision medicine. The authors made a considerable effort in providing summative and comprehensive figures and tables that clearly warrant publication.

It would be good to define the scope of the paper initially as there are many aspects of precision oncology that are not mentioned:

For example:

Include statement that the focus is purely on solid tumour oncology, not hematology

Include a sentence on “precision early cancer detection” and how the “right TIME and right patient” paradigm might shift towards earlier treatment.

Include a sentence on dynamic profiling and that ctDNA makes this possible now: see recent landmark papers in NSCLC and CRC.

Also: the title is confusing. I suggest, to replace “Diversity in Precision oncology” with “Heterogeneity in Precision oncology” in the title and, where required, in the text: 

Diversity in this context is about inclusion of different ethnicity; the series has just also accepted a paper on this subject with diversity in the title.

---

## [Reviewer Report]

Dear Authors,

I have reviewed the revised version of your article “heterogeneity in precision oncology”, which has substantially improved. Table 1 in particular is very interesting in helpful.

One minor comment:

Page 5: “Conversely, in lung cancer, EGFR tyrosine kinase inhibitors are less effective in patients with coexisting TP53” - data are currently limited so I would prefer to concluded “EGFR tyrosine kinase inhibitors might be less effective...”